# Chronic Stress Induces Type 2b Skeletal Muscle Atrophy via the Inhibition of mTORC1 Signaling in Mice

**DOI:** 10.3390/medsci11010019

**Published:** 2023-02-10

**Authors:** Shigeko Fushimi, Tsutomu Nohno, Hironobu Katsuyama

**Affiliations:** Department of Public Health, Kawasaki Medical School, 577 Matsushima, Kurashiki 701-0192, Japan

**Keywords:** chronic stress, soleus muscle, type 2b muscle fiber, REDD1 inhibition, mTORC1 signaling

## Abstract

Chronic stress induces psychological and physiological changes that may have negative sequelae for health and well-being. In this study, the skeletal muscles of male C57BL/6 mice subjected to repetitive water-immersion restraint stress to model chronic stress were examined. In chronically stressed mice, serum corticosterone levels significantly increased, whereas thymus volume and bone mineral density decreased. Further, body weight, skeletal muscle mass, and grip strength were significantly decreased. Histochemical analysis of the soleus muscles revealed a significant decrease in the cross-sectional area of type 2b muscle fibers. Although type 2a fibers also tended to decrease, chronic stress had no impact on type 1 muscle fibers. Chronic stress increased the expression of *REDD1, FoxO1, FoxO3, KLF15, Atrogin1,* and *FKBP5*, but did not affect the expression of *myostatin* or *myogenin*. In contrast, chronic stress resulted in a decrease in p-S6 and p-4E-BP1 levels in the soleus muscle. Taken together, these results indicate that chronic stress promotes muscle atrophy by inhibiting mammalian targets of rapamycin complex 1 activity due to the upregulation of its inhibitor, *REDD1*.

## 1. Introduction

High serum glucocorticoid levels are associated with skeletal muscle atrophy [1], which reduces the rate of protein synthesis and increases the rate of protein breakdown [2]. These conditions include cancer, sepsis, cachexia, AIDS, and protein-calorie starvation. Muscle atrophy also results in weakness, reduced muscle strength, and limited resistance to fatigue. These conditions contribute to progressive disability, reduce the quality of life, and are directly linked to mortality [3]. The mechanisms of glucocorticoid-induced muscle atrophy involve two major proteolytic mechanisms, including the ubiquitin–proteasome and lysosomal systems [1].

Recently, many signaling pathways regulating muscle mass have been identified [4]. There are three major outcomes for skeletal muscle regulation: mitochondrial biogenesis, protein synthesis (hypertrophy), and muscle atrophy. Exercise induces mitochondrial biogenesis via the peroxisome proliferator-activated receptor-gamma coactivator (PGC)-1α1 pathway. Insulin growth factor-I and insulin promote protein synthesis via the phosphoinositide 3-kinase/AKT–mammalian target of rapamycin (mTOR) pathways, whereas myostatin, angiotensin II, and glucocorticoids induce muscle atrophy via autophagy, apoptosis, and the ubiquitin–proteasome system [4]. The total amount of skeletal muscle mass at any time is a direct result of the balance among these signaling pathways. In addition, inhibition of Sirtuin 6, a chromatin-associated deacetylase, leads to maintaining muscle mass against glucocorticoid-induced muscle atrophy through the PI3K/AKT signaling pathway [5]. Moreover, obestatin, which consists of twenty-three amino acids and is called preproghrelin, binds to the G-protein coupled receptor GPR39 to regulate myogenic differentiation [6]. Obestatin signaling targets the KLF15 and FoxO transcription factors and demonstrates an inhibitory effect against glucocorticoid-induced muscle atrophy.

Mammalian skeletal muscles include fast- and slow-twitch muscle fibers [7]. Individual muscle fibers are histologically classified as types 1, 2a, 2b, or 2c based on the nature of their myosin ATPase activity. Concerning the specific characterization, type 1 fibers are the slow-twitch type, type 2a fibers are the fast-twitch type, type 2b fibers are the very fast-twitch type, and type 2c fibers are intermediate, i.e., between the slow- and fast-twitch types. Muscle fiber content differs between the strains of inbred mice. The distribution of muscle fiber types of C57BL/6J strain includes 34% of type 1, 59% of type 2a, 6% of type 2b, and 1% of type 2c [7].

Recent research has focused on the interactions between bone and muscle [8] and how physical load increases bone mineral density (BMD). In contrast, chronic stress induces psychological and physical disorders. Water-immersion restraint stress (WRS) is a well-characterized chronic stress model. WRS in a mouse model results in diminished BMD [9], altered tryptophan catabolism [10], gastric damage [11], and sleep disturbances [12]. However, analyzing the impact of repetitive or chronic stress can be challenging, as inbred mice strains frequently become habituated to a given set of experimental conditions.

The underlying mechanisms of chronic stress-induced muscle atrophy were explored in this study. Male C57BL/6J mice were subjected to repetitive stress, followed by the histological evaluation of fiber types and signaling pathways in the soleus muscles.

## 2. Materials and Methods

### 2.1. Animals and Experimental Protocol

Seven-week-old male C57BL/6J mice were purchased from CLEA Japan (Tokyo, Japan). The mice were housed on a constant humidity and 12 h light/12 h dark cycle in a temperature (22.2 °C)-controlled room and given ad libitum access to food and water. They were acclimated under these conditions for one week. The mice were then divided into two groups: WRS (*n* = 8) and control (*n* = 8) groups. The WRS method has been previously described in detail [9,11,13,14]. Briefly, mice were restrained in a 50-mL conical centrifuge tube with multiple punctures and vertically immersed in the xiphoid process in a 24 °C ± 1 °C water bath for 6 h per day, five times per week, for four weeks. Control mice were maintained under standard conditions. The body weight of each mouse was measured every week, up to and including the final day of WRS. Blood was collected from the lateral tail vein on the second day and weeks two and four to measure serum corticosterone levels. Urine samples were continuously collected for 16 h after the final WRS trial. At the completion of the experiment, mice were anesthetized with sevoflurane, and serum samples were collected using MiniCollect serum tubes (Greiner Bio-One, Kremsmünster, Austria). To determine muscle fiber typing, each group was allocated five mice. All experiments were approved by the Animal Research Committee (#14-046) of Kawasaki Medical School, Japan, in compliance with the ARRIVE guidelines.

### 2.2. Histological Analysis of the Thymus

Histological analysis of the thymus were performed to evaluate the impact of chronic stress. The thymus was fixed with a 10% formalin neutral buffer solution (Wako, Osaka, Japan), embedded in paraffin, sectioned into 5-mm thick sections, and stained with hematoxylin and eosin (H&E). Tissues were evaluated under a light microscope (Olympus).

### 2.3. Measurement of BMD

After sacrificing the mice, the right femurs were collected and fixed in 70% ethanol. Fixed right femurs were evaluated using an X-ray computed tomography for small experimental animals (Model LaTheta, LCT-200; Aloka, Osaka, Japan). Each femur was horizontally placed inside a tube and scanned using a 96-m voxel, as described previously [7]. Total, trabecular, and cortical BMD were measured from the growth plate to 3 mm proximal at the femoral distal end.

### 2.4. Biochemical Markers and Bone Turnover Markers

Serum alkaline phosphatase (ALP), aspartate aminotransferase (AST), lactate dehydrogenase (LDH), creatinine kinase (CK), blood urea nitrogen (BUN) levels, serum creatinine (CRE) levels, and urinary creatinine (U-CRE) levels were measured by Nagahama Life Science Laboratory, Oriental Yeast Co., Ltd. (Shiga, Japan). Plasma corticosterone levels on the second day and weeks two and four were determined using commercially available EIA kits (YK2 40, Yanaihara Ins. Inc., Shizuoka, Japan). Serum levels of Gla-osteocalcin (Gla), a bone formation marker, were determined using a mouse Gla-OC competitive enzyme immunoassay (EIA) kit (MK111, TAKARA Biomedicals, Kyoto, Japan). Bone resorption markers, including serum tartrate-resistant acid phosphatase-5b (TRAP-5b) and urinary C-terminal telopeptides of type I collagen (CTX), were evaluated using a mouse TRAP assay kit (SB-TR103, Immunodiagnostic Systems, Fountain Hills, AZ, USA) and a commercial EIA kit (AC-06F1, Immunodiagnostic Systems, Fountain Hills, AZ, USA), respectively. Urinary CTX levels were corrected by urinary CRE concentrations.

### 2.5. Grip Strength Test

Grip strength was measured when mice were grasping mesh wire with their forelimbs. After the mice were placed on the mesh, they were gently pulled horizontally by their tails until they lost their grip. Peak forelimb grip strength (g) was measured using an automated Grip Strength Meter (MK-380M, Muromachi, Tokyo, Japan) once a week until the last day of WRS.

### 2.6. Measurement of Muscle Volume

Muscle volumes between the two tibia–fibula contact points were analyzed using X-ray computed tomography for small experimental animals (LaTheta LCT 200; Aloka Co., Ltd., Osaka, Japan) with mice maintained under sevoflurane anesthesia. The muscle volume (cm^3^) of each hind limb was calculated by multiplying tissue area (cm^2^) by the slice thickness (96 µm).

### 2.7. Adenosine Triphosphate (ATPase) Staining to Identify Muscle Fiber Type

Soleus muscles frozen were serially cryosectioned at 12 µm in cold acetone to identify individual muscle fiber types. Transverse serial sections in the middle of the soleus muscles were stained with both H&E and ATPase. Muscles were stained with ATPase after pretreatment at pH 4.2, 4.3, 4.4, 4.5, and 10.3 to identify types 1, 2a, 2b, and 2c fibers. The stained sections were photographed at 20× magnification using the All-in-One Fluorescence Microscope (Keyence, Osaka, Japan) to distinguish the fiber types in the central region of the soleus muscle. The ratios of type 1, 2a, 2b, and 2c fibers within each cross-sectional area were determined using the BZ-H3C measurement module (Hybrid Cell Count Vers. 1.1, Keyence).

### 2.8. Quantitative Real-Time Polymerase Chain Reaction (PCR)

The soleus muscle was extracted using the RNeasy Fibrous Tissue Mini Kit (Qiagen, Hilden, Germany). Total RNA was reverse-transcribed into cDNA using the PrimeScript RT Master Mix (Takara Bio, Shiga, Japan). Real-time PCR was performed using the Stratagene Mx3000P Real-Time QPCR System (Agilent Technologies, Santa Clara, CA, USA). The TaqMan probes used in this study are listed in Table 1. The thermal cycling conditions included 1 cycle at 95 °C for 3 min and forty cycles at 95 °C for 10s and 60 °C for 20 s. The relative mRNA expression levels for each gene were normalized by β-actin using the ΔΔ method.

### 2.9. Western Blot Analysis

Soleus muscles were extracted using the EzRIPA Lysis Kit (ATTO, Tokyo, Japan). Protein samples were resolved by electrophoresis in 5–20% polyacrylamide-SDS gel (ATTO) as previously described [15]. After SDS-PAGE, proteins were transferred onto a polyvinylidene difluoride (PVDF) membrane (Merck Millipore, Darmstadt, Germany) and processed further for immunostaining. The membrane was blocked with 5% nonfat milk or PVDF Blocking Reagent for Can Get Signal (TOYOBO, Osaka, Japan) and subsequently incubated with primary antibodies overnight at 4 °C with a horseradish peroxidase-conjugated secondary antibody (HRP-antirabbit IgG, 1:5000, Amersham, GE Healthcare, Little Chatfont, UK) and with ECL Prime Western Blotting Detection Reagent (Amersham, GE Healthcare, UK).

The primary antibodies used in this study were as follows: phospho-S6 ribosomal protein (1:1,000, #4858, Cell Signaling Technology, Danvers, MA, USA), phospho-4E-BP1 (1:1,000, #2855, Cell Signaling Technology), and GAPDH (1:5,000, #5174, Cell Signaling Technology).

Signals were detected using a luminoimage analyzer (ImageQuant LAS4000, GE Healthcare Bio-Sciences, Sweden).

### 2.10. Statistical Analysis

All data are expressed as means ± standard deviation. Statistical analysis were performed using the Student’s *t*-test using the JMP 9 software (SAS Institute Inc., Cary, NC, USA). *p* values of <0.05 were considered statistically significant. We exclude outliers to avoid misleading data.

## 3. Results

### 3.1. Influence of Chronic Stress on Mice Growth

The body weights of mice in the WRS group were lower than those of mice in the control group (Table 2), although WRS had no impact on serum AST, LDH, CK, BUN, and CRE levels. These results suggest that WRS had no immediate impact on liver or kidney function. U-CRE excretion was significantly lower in the WRS group than in the control group. U-CRE is a nitrogenous waste product that is not a protein and is derived from muscle creatinine. Our results suggest that the decreased excretion of U-CRE is related to the decrease in muscle mass in mice in the WRS group.

### 3.2. Influence of Chronic Stress on Corticosterone Levels, Thymus Mass, and BMD

Corticosterone level and thymus mass were measured as previous studies suggested that chronic stress might have a direct impact on this organ [16,17]. Plasma corticosterone levels on the second day and weeks two and four were significantly higher in the WRS group than in the control group (Table 2). However, plasma corticosterone levels gradually decreased in both groups, suggesting that the mice become accustomed to WRS. The corticosterone level on day zero did not measure because the mice were not accustomed to the circumstances. Additionally, corticosterone level was compared after stress conditions. At week 4, the thymuses were dramatically smaller in mice in the WRS group than in those in the control group (Figure 1A). Histological analysis revealed that the thymus in the WRS group strongly degenerated. Furthermore, the border of the thymic cortex could not be distinctly detected (Figure 1B).

A previous study has revealed that WRS has a measurable impact on BMD [9]. Therefore, BMD and bone turnover markers were examined. Total, trabecular, and cortical BMDs were all significantly lower in the WRS group than in the control group (Table 3). Comparing the WRS group and the control group for markers of bone formation, we found no difference in ALP, but carboxylated (Gla) levels were significantly lower in the WRS group than in the control group. Concerning bone resorption markers, urinary CTX and serum TRAP5b levels were significantly higher in mice in the WRS group than in those in the control group.

Taken together, these results suggest that mice exhibit a stress response and osteopenia after a four-week WRS trial.

### 3.3. Impact of Chronic Stress on Hind Limb Muscle Size and Muscle Strength

The effect of WRS on hind limb muscles was examined. Muscle mass was determined cross-sectionally via computed tomography (CT) at week 4 of WRS. WRS resulted in significantly decreased muscle size (Figure 2A,B). A forelimb grip strength test was performed to test muscle strength. WRS also resulted in a significant decrease in muscle strength (Figure 2C). These results indicated that WRS has an impact on muscle size and strength.

### 3.4. Influence of Chronic Stress on the Soleus Muscle

Soleus muscles were subjected to morphological and histological analysis. The soleus muscle contains slow-twitch (type 1) and fast-twitch (type 2) fibers; fast-twitch fibers can be further categorized into types 2a, 2b, and 2c. ATPase staining at different pHs was used to identify these muscle fibers (Figure 3). Since all the fibers from the muscle of mice in the WRS group were thinner than those in the control group, we measured the area rather than counting the number of each fiber.

Although WRS did not affect the fiber types ratio, it resulted in a 15% reduction in the cross-sectional area of type 2b muscle fiber compared to control mice (Table 4). These results indicated that WRS significantly decreased the size of type 2b fibers in the soleus muscles.

### 3.5. Molecular Responses to Chronic Stress within the Soleus Muscle

Since the plasma levels of corticosterone were higher in the WRS group, we evaluated the expression of glucocorticoid-responsive genes in the muscle tissue. The expression of *FoxO1, FoxO3, KLF15, Atrogin-1, REDD1*, and *KFBP5* was significantly higher in the muscle tissue of mice in the WRS group. However, this did not affect the expression of *myostatin* or *myogenin* (Figure 4A). Likewise, we observed a tenfold increase in the expression of *REDD1* in mice in the WRS group. Therefore, REDD1 signal transduction was evaluated. REDD1 is a negative regulator of mTORC1 signaling. The activation of S6K1 and 4E-BP1 was determined via western blot analysis. Immunoreactive p-S6K1 and p-4E-BP1 levels were significantly decreased in the soleus muscle at week 4 in the WRS group (Figure 4B,C). Decreased expression levels were also detected in the tibialis anterior, which is a fast-twitch glycolytic muscle (data not shown).

## 4. Discussion

To explore the relationship between chronic stress and muscle mass, we performed experiments in which mice were subjected to four weeks of chronic stress using WRS. Chronic stress resulted in decreased body weight, a decrease in the size of the hind limb muscle, and a decrease in U-CRE levels, indicating that chronic stress induces muscle atrophy. Histological analysis revealed the atrophy of type 2b skeletal muscle fibers in response to chronic stress.

Skeletal muscle mass is regulated via multiple signal transduction pathways. Corticosterone binding to its receptor activates REDD1, a negative regulator of mTORC1 [18]. Since *REDD1* expression was ten times higher in the WRS group than in the control group, we investigated the expression levels of proteins downstream of REDD1. As mTORC1 signaling promotes protein synthesis via the phosphorylation of p70S6 kinase 1 (S6K1) and inhibits protein synthesis via the phosphorylation of eIF4E-binding protein (4EBP) [19], mTORC1 activation is typically associated with muscle hypertrophy [20]. Moreover, the phosphorylation of S6K1 plays a key role in muscle protein synthesis [1]. In the present study, chronic stress-activated REDD1 expression and inhibited the phosphorylation of S6K1 and 4E-BP1 in the soleus muscle. These results indicate that the phosphorylation of S6K1 and the mTOR signaling pathway could be inhibited by chronic stress, which could cause muscle atrophy. Moreover, corticosterone-induced muscle atrophy is observed in fast-twitch glycolytic muscles, such as the gastrocnemius muscle, but not in the soleus muscle [21,22]. Even though the cross-sectional area of type 1 slow-twitch muscle fibers did not change, the cross-sectional areas of type 2b very fast-twitch fibers were smaller in chronically stressed mice than in control mice. Since the soleus muscle contains both fast-twitch and slow-twitch muscles [23,24], we used the soleus muscle in this study. Stress-induced muscle atrophy was first observed in very fast-twitch muscle fibers in the soleus muscle. Mechanisms underlying the difference among muscle fiber types between type 1 and type 2 were not clear in this study.

Other than the mTORC1 signaling pathway, FoxO, KLF15, Atrogin-1, myogenin, and myostatin are all associated with muscle protein degradation [16]. FoxO is a subfamily of the forkhead-type transcription factor family [25], and FOXO1 may suppress an increase in muscle mass, notably of type 1 muscle fibers. Moreover, FoxO is an upstream regulator of both Atrogin-1 and MuRf-1, whereas KLF15 is an upstream regulator of FoxO [18]. In the present study, these genes were all upregulated in chronically stressed mice, although their expression levels were lower than those of *REDD1*. Based on these results, these pathways could also be associated with protein degradation and muscle mass loss. In contrast, *myostatin* and *myogenin* levels were lower in chronically stressed mice than in control mice. These findings suggest that the autocrine- or paracrine-like effects of *myostatin* and *myogenin* did not play a role in promoting these observations. Myostatin is a member of the transforming growth factor-b family that is mainly expressed in the skeletal muscle [26] and is associated with protein degradation at this locale. Body and muscle mass are decreased, and myostatin expression is increased in response to daily psychological stresses, such as restraints or cage switching [27]. Myogenin is a transcription factor that regulates muscle size; as such, mouse models lacking myogenin have reduced body mass and muscle fiber size [28]. Taken together, myostatin and myogenin do not appear to be associated with chronic stress, at least for WRS.

Male mice are typically used in chronic stress models as they become endocrinologically stable in adolescence [9,10,11,12,13,14]. The male C57BL/6J mice featured in this study responded to WRS with a decrease in thymus size and an increase in plasma corticosterone levels. These results demonstrate that WRS can be successfully used to construct a chronic stress model with no evidence of adaptation or habituation. Repeatedly stressed mice show severe loss of lean body mass, hyperglycemia, and dyslipidemia associated with hypercortisolism, hyperleptinemia, and insulin resistance [29]. When considering parallel human cases, twenty-eight days of bed rest can result in a 28.4% loss in leg extension strength, a threefold loss in lean leg mass, and a twofold increase in plasma cortisol levels [30]. These results suggest that repeated stress or inactivity results in loss of body weight and muscle mass. In the present study, loss of muscle mass may also be related to prolonged periods of inactivity. In addition, elevated corticosterone levels were observed, which might contribute to muscle atrophy.

A part of our study also focused on changes in bone–muscle interactions in response to chronic stress. Osteocalcin secreted from osteoblasts promotes the exercise capacity of mice [31]. When osteocalcin binds to its receptor (Gprc6a), it induces an increase in nutrient uptake and catabolism of myofibers. In the present study, we observe a decrease in Gla, a marker of bone formation, and an increase in the bone resorption markers CTX and TRACP-5b. These factors might be associated with skeletal muscle loss.

This study has several limitations. First, even though we designed the study to address the role of chronic stress in promoting skeletal muscle atrophy, we cannot rule out the possibility that long periods of inactivity associated with WRS might also be involved. Second, mRNA was extracted from the whole soleus muscle. Therefore, we were unable to draw any conclusions regarding gene expression in specific muscle fiber types. Third, we could not set the time course of muscle atrophy. To properly observe muscle atrophy, many time points should be considered. Transgenic or knockout mice should be used in future research to better understand the mechanism of muscle atrophy.

In conclusion, chronically stressed mice experience a significant decrease in the cross-sectional area of type 2b skeletal muscle fibers in association with the activation of the mTORC1 signaling pathway. Further studies should be performed to clarify the unique gene expression patterns of each muscle fiber type.

## Figures and Tables

**Figure 1 medsci-11-00019-f001:**
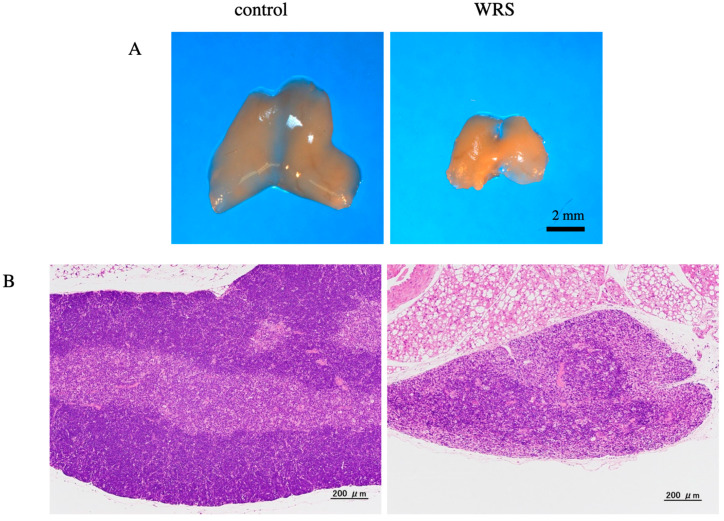
Morphological evaluation of the thymus. Thymus size and histological differences were compared between the control and WRS groups. (**A**) Thymus size was dramatically decreased in mice in the WRS group. (**B**) Histological analysis revealed atrophy of the thymic cortex in mice in the WRS group.

**Figure 2 medsci-11-00019-f002:**
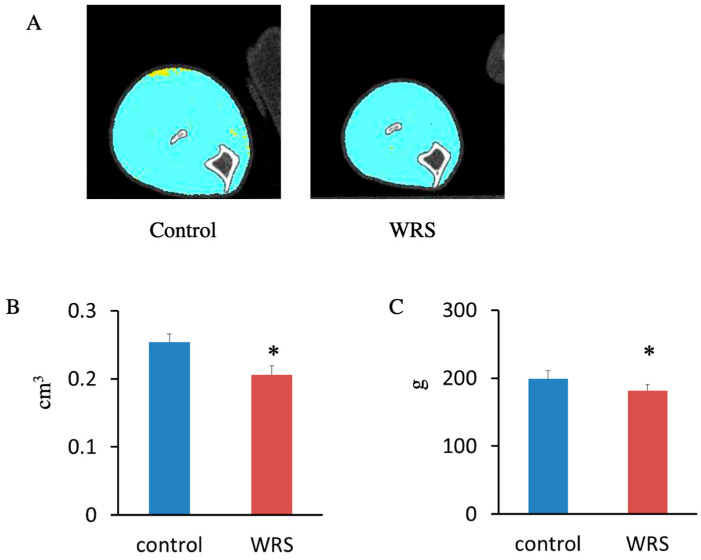
Cross-sectional evaluation of hind limb and muscle strength testing. (**A**) Hind limb muscle size was evaluated via computed tomography. (**B**) The size of the hind limb muscle was significantly decreased in the WRS group. (**C**) Grip strength in the WRS group was significantly lower than that in the control group. The sample size was *n* = 8 for the control and WRS groups. *: *p* < 0.05.

**Figure 3 medsci-11-00019-f003:**
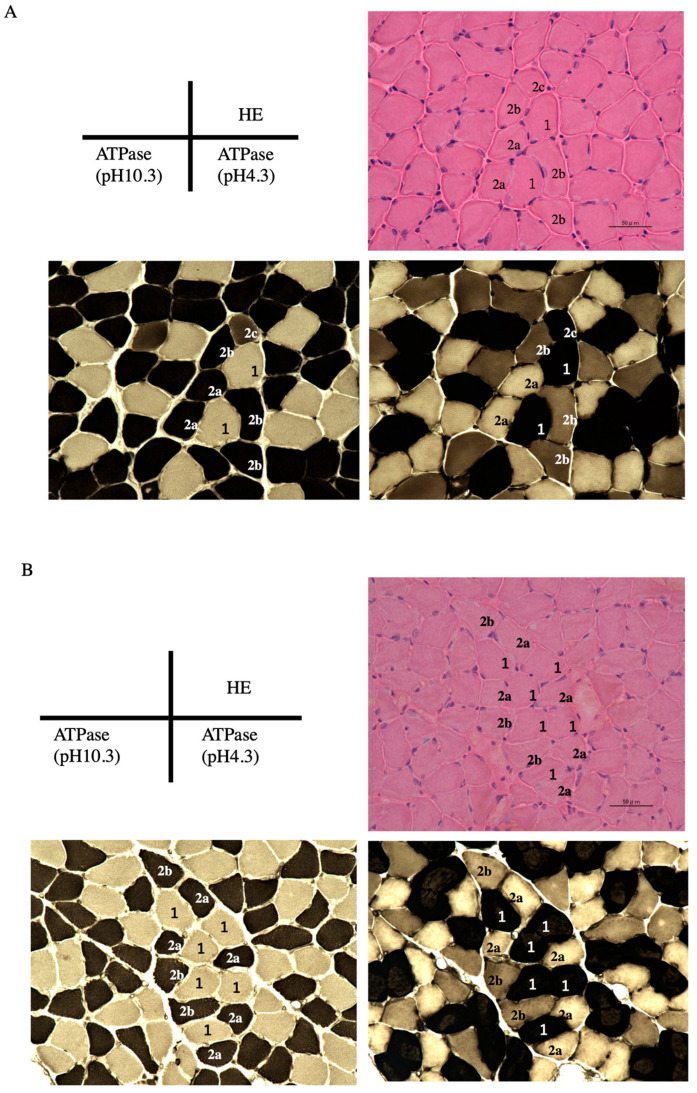
Histological analysis of the soleus muscle. Muscle fiber typing was performed by determining myosin ATPase activity; ATPase activity at pH 4.3 is associated with type 1 fibers and ATPase activity at pH 10.3 is associated with type 2 fibers. (**A**) Control group; (**B**) WRS group. The muscle fibers in the WRS group had thinner cross sections and contained fewer type 2b fibers than those in the control group. The scale bar indicates 50 mm.

**Figure 4 medsci-11-00019-f004:**
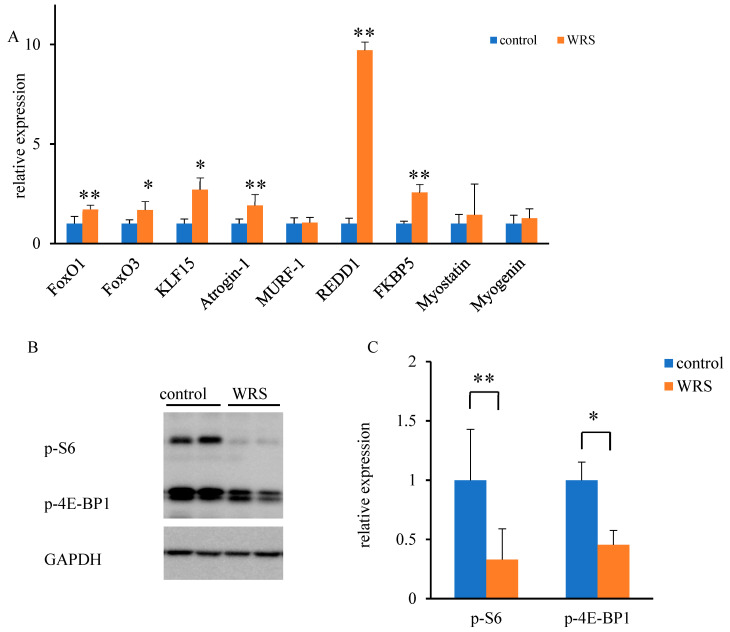
Expression of glucocorticoid-responsive genes under chronic stress conditions. (**A**) Expression levels of genes associated with protein degradation were significantly higher in the WRS group, although myostatin, an inhibitor of muscle proliferation, and myogenin, a marker of muscle differentiation, remained unaffected. (**B**,**C**) Phosphorylation of S6 and 4E-BP1, both downstream regulators of mTOR-mediated signal transduction, was significantly lower in mice in the WRS group than in those in the control group. **: *p* < 0.01, *: *p* < 0.05. Original picture for Western blot analysis in all mice is shown in Appendix A.

**Table 1 medsci-11-00019-t001:** Taqman probes were used in the present study.

Gene Name	Assay ID
FoxO1	Mm00490672_m1
FoxO3	Mm01185722_m1
KLF15	Mm00517792_m1
Atrogin-1	Mm00499523_m1
MuRF1	Mm01185221_m1
REDD1	Mm00512504_g1
FKBP5	Mm00487406_m1
Myostatin	Mm01254559_m1
Myogenin	Mm00446195_g1
GAPDH	Mm99999915_g1

**Table 2 medsci-11-00019-t002:** Influence of WRS on mice growth.

	Control	WRS
body weight (g)	25.2 ± 0.3	24.0 ± 0.4 *
urinary creatinine (U-CRE) (mg/dL)	21.43 ± 4.78	8.67 ± 3.18 **
aspartate aminotransferase (AST) (IU/L)	155.0 ± 34.2	192.0 ± 53.6
lactate dehydrogenase (LDH) (IU/L)	647 ± 190	753 ± 290
creatinine kinase (CK) (IU/L)	1758 ± 715	1847 ± 889
blood urea nitrogen (BUN) (mg/dL)	36.8 ± 2.0	38.8 ± 8.1
serum creatinine (CRE) (mg/dL)	0.13 ± 0.02	0.14 ± 0.10
corticosterone level (ng/mL)		
day 2	168.1 ± 65.1	370.6 ± 44.1 **
week 2	71.6 ± 36.8	295.0 ± 61.4 **
week 4	82.4 ± 34.6	249.4 ± 45.8 **

Values were expressed as mean ± SD, and Student’s *t*-test was performed. *: *p* < 0.05, **: *p* < 0.01.

**Table 3 medsci-11-00019-t003:** Influence of WRS on BMD and bone turnover markers.

	Control, *n* = 5	WRS, *n* = 8
total BMD (mg/cm^3^)	391 ± 22	328 ± 29 **
trabecular BMD (mg/cm^3^)	188 ± 15	146 ± 30 *
cortical BMD (mg/cm^3^)	1052 ± 55	982 ± 32 *
ALP (IU/L)	329.0 ± 26.3	402.3 ± 70.9
Gla (ng/mL)	25.8 ± 6.2	11.3 ± 3.5 **
CTX (ng/mM Cre)	6.1 ± 0.7	26.9 ± 7.9 **
TRACP-5b (U/L)	12.5 ± 1.1	16.4 ± 3.0 *

Values were expressed as mean ± SD, and Student’s *t*-test was performed. *: *p* < 0.05, **: *p* < 0.01.

**Table 4 medsci-11-00019-t004:** Muscle fiber typing and cross-sectional area of soleus.

	Control, *n* = 5	WRS, *n* = 8
ratio of fiber type (%)		
Type 1	34.0 ± 4.3	35.4 ± 4.3
Type 2a	46.9 ± 5.2	44.8 ± 2.1
Type 2b	17.4 ± 3.0	15.8 ± 8.7
Type 2c	1.2 ± 1.1	3.4 ± 5.5
area of fiber type (mm^2^)		
Type 1	1312 ± 156	1265 ± 172
Type 2a	1135 ± 115	990 ± 85 #
Type 2b	1296 ± 152	1084 ± 97 *
Type 2c	836 ± 446	532 ± 467

Ratio and areas of fiber type in soleus muscle by myosin ATPase activity were investigated. Cross-sectional area of Type 2b fiber was significantly smaller in the WRS group. Values were expressed as mean ± SD, and Student’s t-test was performed. *: *p* < 0.05 #: *p* < 0.1.

## Data Availability

The data presented in this study are available upon request from the corresponding author.

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
