# Peer review of "Chronic Stress Induces Type 2b Skeletal Muscle Atrophy via the Inhibition of mTORC1 Signaling in Mice"

_medsci, 2023, doi:10.3390/medsci11010019_

Round 1

Reviewer 1 Report

The authors conclude that glucocorticoid-induced skeletal muscle atrophy affects type 2 fibers while leaving type 1 fibers unaffected. However, since there is no explanation for the mechanism by which fiber specificity arises. Authors should explain the mechanism underlying occurrences the difference among muscle fiber type.  Also, authors cited in the manuscript that glucocorticoid-induced skeletal muscle atrophy is observed in fast muscles such as the gastrocnemius muscle but not in slow muscles such as the soleus muscle.  (Lines 264-265). However, the authors used the soleus muscle in this study. Authors should explain the mechanism by which slow-twitch versus fast-twitch skeletal muscle atrophy is different.

Need spacing between “transforming growth factor-b” and “family”

Author Response

Thank you very much for your useful comments. Our responses to your comments were as follows.

Unfortunately, we could not explain precise mechanisms of both type 1 and type 2 muscle atrophy, we described this point in line 452-454. In addition, we described why we used soleus muscle in line 450-451.

We added space between 'transforming growth factor-b' and 'family'.

Reviewer 2 Report

Please find attachment.

Author Response

Thank you very much for your useful comments. We amended our manuscript as follows.

  1. We added more recent and relevant literatures in Introduction (line42-48).
  2. In Table 2, we could not add data for day 0, so we added this point in line 192-194.
  3. We added sample size in legend of Figure 2.
  4. It is hard to correct scale bar in Figure 3 since original figure contain this bar. Instead of correction, we added ‘scale bar indicates 50 mm’ in legend.
  5. We recalculated the relative expression of MuRF1 and p-S6 after exclusion of outlier. Statistical difference of MuRF1 was disappear, but still statistical difference was observed in p-S6. We rewrite manuscript considering this point.

Reviewer 3 Report

Shigeko Fushimi et al report that Chronic Stress Induces Type 2b Skeletal Muscle Atrophy via 2 the Inhibition of mTORC1 Signaling in Mice. This study is potentially interesting. However, there are several concerns listed below. This paper will be strengthened by addressing the following issues.

1.    The authors showed that chronic stress has some effect (muscle fiber type and molecular response) on the soleus muscle. In the soleus muscle, chronic stress decreased area of type 2a/2b but not type I. Soleus muscle is a representative muscle of type I. Therefore, the authors should add data of TA (fast-twitch muscle) muscle. I mean, because WRS has some effect on type 2 muscle (muscle fiber type and molecular response), you should check some effects on TA muscle. Although, you showed molecular response of WRS on soleus and TA muscle, you insist that WRS induces Type 2b muscle atrophy. I think that this is a little bit confusing and doesn’t make sense.

2.    I wonder what is the most important regulator in the effects of WRS? (among the so many targets, ex, REDD1, Foxo1, Foxo3, KLF15, Atrogin, MuRF1 and FKBP5). And the authors need to include rescue study for the important regulator. If so, the authors could suggest solid mechanism for WRS. Because this study is animal study, it is not easy to use TG or KO mice. If so, they should use skeletal muscle cell with knock down and overexpression technique.

Author Response

Thank you very much for your useful comments. We amended our manuscript as follows.

  1. We added the reason why we used soleus muscle in line 450-451.
  2. We concluded that REDD1 was the most important regulator expressed in line 429-447. In addition, we added for the need of usage of transgenic or knockout mice for further study in line 511-512.

Round 2

Reviewer 1 Report

The authors' manuscript has many unresolved issues, but the story is clear and satisfying, and the reviewers consider it worthy of publication.

Reviewer 3 Report

I support this paper is suitable for publication in Medical Sciences.